# A Mimic Model Approach for Impact Assessment of Mining Activities on Sustainable Development Indicators

**Hesam Dehghani [1], Marc Bascompta [2] , Ali Asghar Khajevandi [3],\* and Kiana Afshar Farnia [1]**

1.  Department of Mine Engineering, Hamedan University of Technology, Hamedan 65155, Iran
2.  Department of Mining, Industrial and ICT Engineering, Polytechnic University of Catalonia, 08034 Barcelona, Spain
3.  Occupational Health and Safety Department, Kashan University of Medical Sciences, Isfahan 73441, Iran
\*   Correspondence: khajevandi.asghar2008@gmail.com

**Abstract:** Mining activities are usually associated with negative outcomes. Therefore, it is crucial to identify and assess these outcomes by the mining company to achieve proper management. The present study has been defined to discover the outcomes of mining activities and their testing in one of the open pit mines of Iran. The present research has been defined into two sections, qualitative and quantitative. The corresponding data of the qualitative section were derived through analysis of the hidden contents of semi-structured interviews with experts and a review of the literature using the Maxqda 2022 software in the forms of open coding and axial coding. In the quantitative section of the study, data were collected via the standard questionnaire and analyzed using the SPSS26 and Mplus software. By coding the interviews and existing documents, 62 primary codes were extracted and classified into 5 main criteria (environmental, health, social, economic, and cultural) in the form of axial coding. The analysis results of the collected questionnaires showed that mining activities had the highest impact on the environment (86.32) and individual health (80.86), while the lower impact was on their economic situation (54.55). The findings of this study showed that there is a significant difference between men and women in terms of the environmental ($p = 0.013$) and economic ($p = 0.01$) indicators. While men believed that the mining activity had caused permanent environmental impacts on their living area, women recognized the mining activities as the cause of economic weakness in their families. Results from the present study could be effective in formulating the controlling strategies for potential negative outcomes of mining and achieving effective sustainable development.

**Keywords:** sustainable development; mining activities; mixed study; coding

## 1. Introduction

Mines are considered one of the most important elements regarding sustainable development [1]. Revenues from mineral resources are used to support economic development [2] and improvement of welfare in some of the most developed countries, such as Australia [3]. For instance, the mining industry have created 634,000 direct jobs and 1,270,000 indirect jobs in the USA in 2012 [4]. While mining products represent about 30–60% of the total export in low to middle-income countries, they are also an important source of foreign investment, reaching 60–90% in Zambia country of Africa [5]. Despite its potential positive economic and social impacts, there are also potential negative elements such as environmental or health and safety issues [6]. Hence, considering the positive and negative outcomes of the sector, the global mining initiative was established by some of the main mining companies in 1998, with the idea of achieving sustainable development and the solutions required to achieve it [7,8].

Sustainable development is generally a combination of social, economic, and environmental dimensions, providing the present needs of human communities without endangering the capabilities of future generations [9]. Thus, mining could well achieve the

goals but could challenge the capabilities of the future generation. These huge economic resources can cause negative outcomes in terms of economy and sustainable environment in their coverage areas. For example, about 900 hectares of the best agricultural lands in Iran have been devastated due to the accumulation of mineral materials and other activities [10]. Continuous reduction in the environmental quality of mining areas threatens the life of residents, becoming a severe social and environmental problem [11]. The mining industry is responsible for many social and environmental effects, including air and water pollution, fear of land tenure, loss of biodiversity, socio-economic disruption [12–16], and local cultural identity dislocation [17–19]. One of the most important indicators of the social dimension, in a mining area, is the migration movements and demographic changes [20,21]. Erwana et al. (2015) analyzed the impact of mining activities on sustainable development in Indonesia, revealing that mining has a direct and negative impact on the society and environment, while a potentially positive impact on the local economy [22]. Besides, Mojarradi et al. (2016) show that mining has also a significant effect on cultural factors such as local accent, style of traditional architecture, change in the lifestyle of indigenous people, rate of crime, and other social deviations [23]. Li et al. (2017) performed a comparative study concerning the impact of mining on the sustainable development indicators in villages different distances from a coal mine, proving that individuals who lived close to mining areas had dissatisfaction with air quality, access to water, and price inflation of goods and services. While people living away from the same coal mines were unsatisfied with welfare factors related to inflation, the price of real estate, and education [24]. Farahani and Bayazidi (2018) reported that the mining activities had caused the drying up of the river or slowing down of its flow, loss of plants around the river, reduction in the animal population, traffic problems, and, consequently, disturbance of life in communities close to the mines [25]. In addition, Yang and Ho (2018) performed a study using the model of behavior modification theory which was carried out on residents of 37 villages near the mining areas in China. The results showed that 77% of respondents believed that pollution in the mining areas is serious and that the government and authorities do not support environmental activities [26]. Kumah (2006) reported that open pit mines, especially gold mines due to their intrinsic characteristics, have significant negative effects and most of the gold mining companies in the developing world are responsible for various environmental, social, and economic issues, such as acid mine drainage, noise, dust, air pollution and water contamination due to presence of arsenic, cyanide, and mercury, resulting in diseases among people, loss of vegetative cover and loss of livelihood and mass displacement [27]. Leuenberger et al. (2018) showed that, in addition to environmental concerns, other impacts such as cracks in houses and extreme noise due to blasting causes concern among individuals, especially women and children as they spend more time at home, even experiencing experienced fear and anxiety in the case of children [28].

The mining sector is very relevant in the Iranian economy, where up to 100,000 individuals are directly involved in the sector. Besides, most of its mines, 97%, are open pit mines. This method of extraction presents higher impacts regarding social, cultural, and environmental issues, compared to underground mining [29]. Various studies have been carried out considering the impacts of mining activity on sustainable development indicators (environment, social, economic, cultural, and health). Accurate identification of these outcomes could greatly help with reducing and controlling them and achieving sustainable development in mining. Assessing the different impacts of mining activities on the local and regional ecosystem and identifying and prioritizing the concerns of people residing around mines. The present research aims to explore the components of sustainable development in open pit mining, based on the assessment of Iranian mining activities.

Further, the study contributes to the literature in several ways:

First, to the best of our knowledge, this is one of the first studies that explored and tested sustainable development indicators in the Iran surface mine sector in the form of a mixed study. Due to the limitations of previous studies related to indicators, this study has significant theoretical contributions.

Second, the results of this study are presented in the form of a mimic model which evaluates sustainable development indicators in different groups (gender, education, etc.) in the form of visual analysis.

Third, the study also has some practical implications that instead of dealing with other less important factors, mining managers can identify and promote certain potential critical success factors.

## 2. Materials and Methods

The present research is mixed research, including qualitative and quantitative sections. In the qualitative section, the index and components of sustainable development related to mining were extracted in the form of a phenomenology strategy. While in the quantitative section, the indicators were assessed in terms of their differences considering the standpoint of villagers residing near one of the open pit mines. Figure 1 shows the strategy applied in the present study.

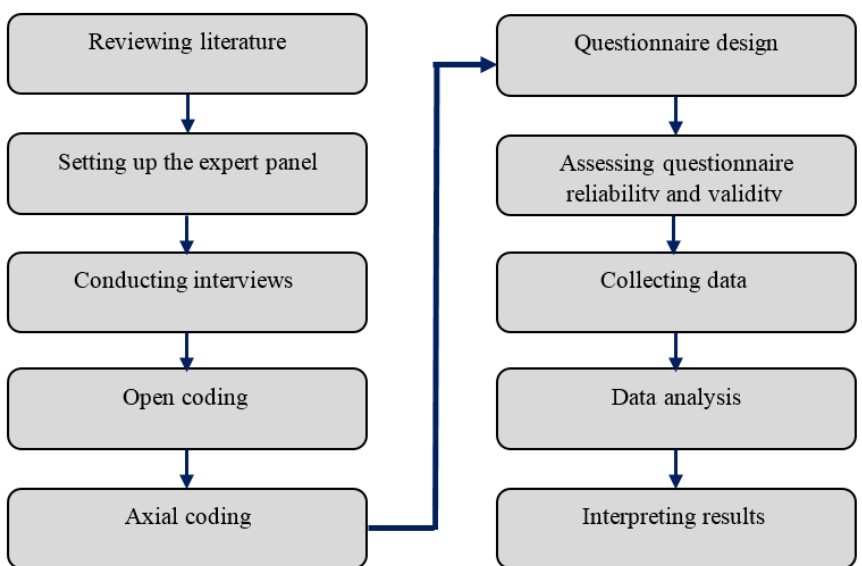

**Figure 1.** Steps followed in the study.

### 2.1. Qualitative Study

Data collection was performed by investigating the texts and semi-structured interviews with experts in the mining field. The semi-structured nature of the interviews makes it possible for the researcher to adjust the interviews with the responses of interviewees and ask secondary and interpretive questions [30]. Thus, the interviewees first filled out the informed consent forms and were informed that they could exit the interview at any time. The researcher applied the probability and purposeful sampling method for selecting the experts [31,32] In the present study, the theoretical saturation approach was used to end the interview; in this approach, the researcher immediately analyzed the interviews and performed the next one based on the output of the previous interview [33–35]. In this regard, the researcher attained theoretical saturation after 12 interviews. Each interview took 50 min to one hour and, after implementing the interviews in the text format, the Maqam software v.2.22 was used for the data analysis, based on the Strauss- Corbin approach. This approach includes the open, axial, and selective coding types [36]. In the open type coding, the primary codes are extracted from the semantic units through analysis of the hidden contents. At the axial coding stage, the codes with common semantics are classified as categories and, finally, those categories will form the subcategories of a main (core) category called sustainable development [37–39]. The primary codes have formed the core of a distributable questionnaire among the research statistical sample. At the next stage, the questionnaire, obtained from a qualitative approach with a 5-point Likert

scale, was assessed in terms of validity and reliability. Initially, the face validity of the questions was investigated [40] and, subsequently, the content validity was examined. For the calculation of the Content Validity Index (CVI), the Waltz and Basel method [41] was implemented, while the Lawshe method was employed [42] to determine the Content Validity Ratio (CVR). Furthermore, Cronbach's alpha coefficient was calculated for each of the identified factors to investigate the reliability. For this purpose, the questionnaire was completed by 20 experts and the data were analyzed using the SPSS software. A Cronbach's alpha coefficient value greater than 0.7 shows its internal consistency for the questions corresponding to each variable [43].

### 2.2. Quantitative Study

In the quantitative section of the study, field data were collected for the representative sample of the population, where its size was determined using G-POWER software [44]. The conditions for entering the study included having a minimum one-year record of living in the studied village, lack of underlying diseases, and filling out the informed consent form. The sample size was calculated equal to 102 individuals considering the probability error of 0.05, the Power of a test (85%), and the effect size of 0.15 (Figure 2).

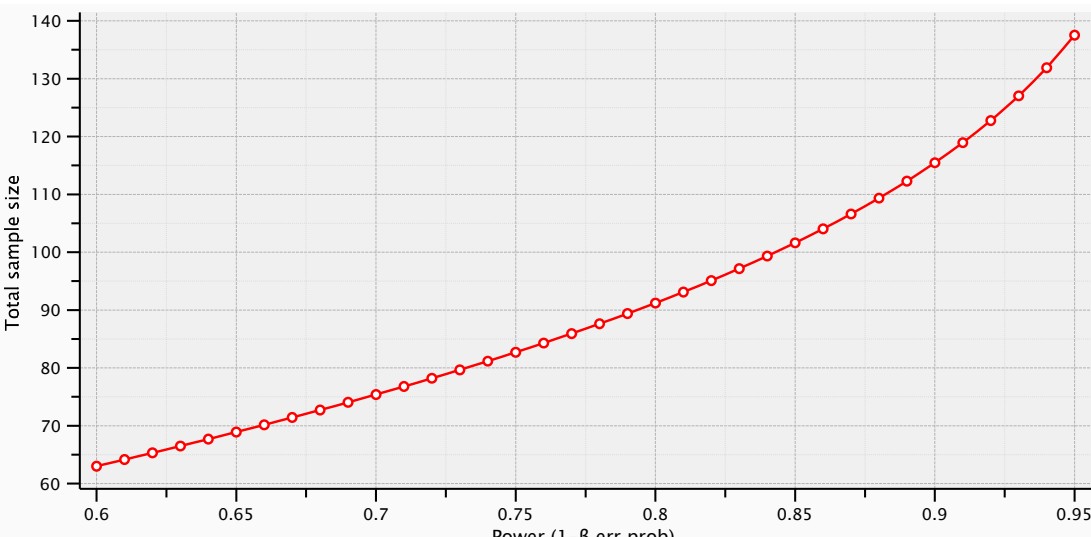

**Figure 2.** The sample size was calculated using the G-POWER software.

After collecting the questionnaires, the sustainable development index for the studied population was calculated using the two SPSS26 and MPLUS 7.4 software, measuring the differences as well.

## 3. Results and Discussion

### 3.1. Qualitative Research Findings

The present study has been carried out with the participation of 12 experts in the fields of mining and environment.

Table 1 shows the demographic information of participants in the present study. After the primary coding of the interviews and determining the significant units, a total number of 62 primary codes were extracted. The frequency of the extracted codes was demonstrated as a cloud diagram (Figure 3). The primary codes included road accidents, sleep disorder, and noise with six repetitions, which had the highest importance. While dust, lifestyle changes, the safety of properties and assets, change in management of the village, and job satisfaction had the minimum importance among the extracted codes, with only one repetition.

**Table 1.** Demographic variables of the experts.

| Demographic Variables | Total | Percentage |
|---|---|---|
| Gender | | |
|     Male | 9 | 75.00% |
|     Female | 3 | 25.00% |
| Educational | | |
|     Bachelor | 1 | 0.08% |
|     Master | 6 | 50.00% |
|     Doctoral | 5 | 42.00% |
| Experience in mine design | | |
|     5–15 years | 7 | 58.33% |
|     >15 years | 5 | 41.67% |

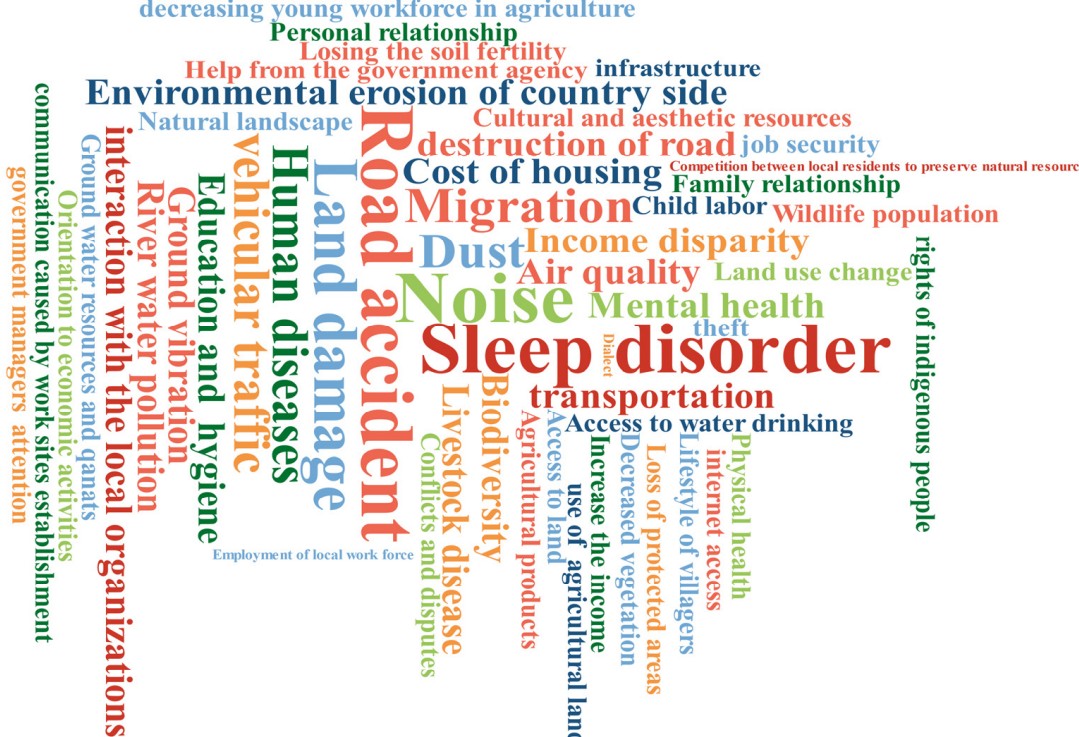

**Figure 3.** Cloud graph representing primary extracted code from expert interviews.

The primary codes were classified using the two methods, either by creating new codes or using the centralized codes, which resulted in exploring five categories concerning the structure of sustainable development in mines. The identified categories consisted of environmental, economic, social, cultural, and health factors. At the next stage, using the initially identified codes, the intended questionnaire was designed. The face validity was investigated by the research team and, then, it was asked from 10 experts to express their opinion on the necessity of each question in the questionnaire for assessing each category. Subsequently, the corresponding CVR of each question was calculated by Equation (1) and the questions with CVR values greater than the acceptable value (0.62 for 10 experts) remained. The formula of the content validity ratio is

$$\mathrm{CVR} = \frac{N_e - N/2}{N/2} \tag{1}$$

in which Ne is the number of panellists indicating "essential" and N is the total number of panelists.

$$CVI = \frac{N_p}{N_t} \tag{2}$$

In the above relationship, CVI is the validity index, Np is the number of experts that gave three and four points to the item and Nt is the total number of experts. According to the proposed criterion by Lynn [45], where the number of experts is between six and ten, the minimum acceptable CVI value is equal to 0.78. Considering that the number of experts in the current study was 10, thus the questions where their CVI value was less than 0.78 were removed, with a total of five questions removed. Table 2 shows the reliability results of the questions corresponding to each variable in the questionnaire.

**Table 2.** Cronbach's alpha coefficient corresponding to the research variables.

| Variables | No. of Indicators | Cronbach's Alpha Value |
| --- | --- | --- |
| Environmental impact | Q1–Q17 | 0.78 |
| Economic impact | Q18–Q26 | 0.83 |
| Social impact | Q27–Q47 | 0.76 |
| Cultural impact | Q48–Q50 | 0.81 |
| Health impact | Q51–Q55 | 0.78 |

*3.2. Quantitative Research Findings*

The questionnaires were delivered to individuals able to read and write. For those who were illiterate, the questions were asked and their responses were recorded in the questionnaire. According to frequencies observed in the sample members, 40.2% (41 individuals) were women, and 59.8% (61 individuals) were men. Most of the participants were illiterate (31%) and only 5% had a higher level than a diploma (Bachelor's degree). 82.4% (84 individuals) were married and 17.6% (18 individuals) were single. 58.9% of the participants were older than 40 years. About 97% of the respondents had a residence record higher than 5 years.

According to the results displayed in Figure 4, the mean score of expressed opinions by the women, in the studied sample, is higher than the men's perspective in terms of health, cultural, social, and economic outcomes due to the mining activities. On the other hand, the mean score of men is higher than women in terms of the environmental dimension. Men mostly believe that mining activities have affected the environment of their living area.

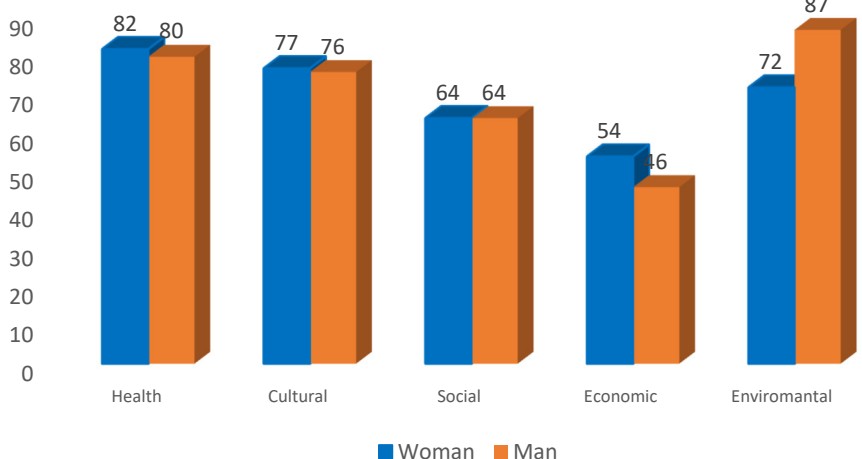

**Figure 4.** Comparison between the mean of scores of sustainable development indicators in terms of gender in the studied sample.

Mean scores of sustainable development indicators in the studied sample, Figure 5, show that individuals with higher education mostly believe that mining activities have an impact on their environment. They also believe that their economic condition has gotten worse. The mean health index is higher in the illiterate or with a low level of literacy individuals than the literate ones. It means that the illiterate individuals further agreed upon this issue that mining activities have adversely affected their health.

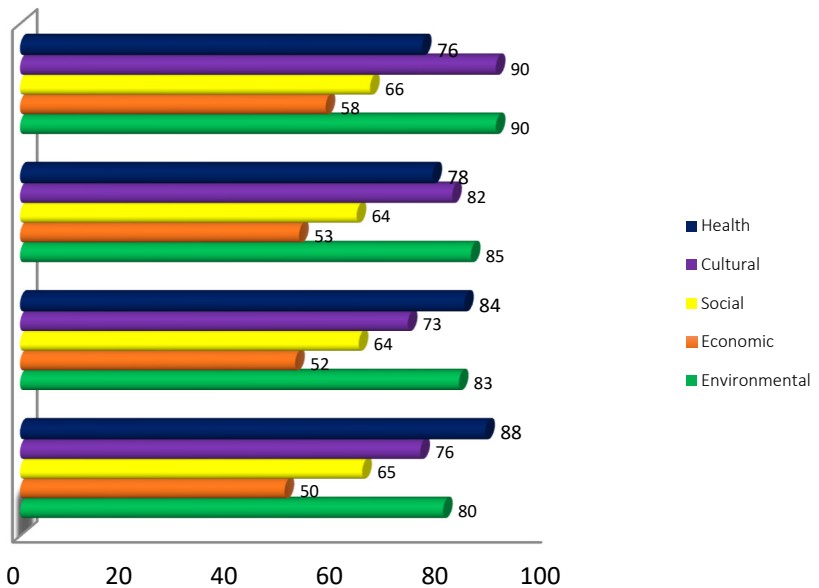

**Figure 5.** Comparison between the mean scores of sustainable development indicators in terms of the education level.

According to Figure 6, the mean score of the individuals with greater records of residence in areas close to the mines shows a higher value for environmental, health, and social indicators in comparison to individuals with lower records of residence. On the other hand, the outcomes in the economic index give opposite results, individuals with lower records of residence feel less about the negative economic outcomes. On the other hand, the cultural outcomes due to mining activities for the individuals with residence records between 10 and 15 years are greater concerning the two other groups.

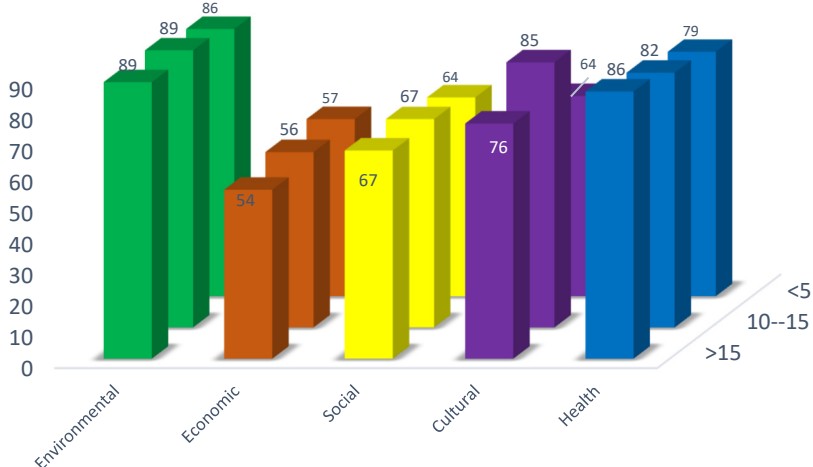

**Figure 6.** Comparison between the mean of scores of sustainable development indicators in terms of residence records.

Comparing the mean scores of sustainable development indicators in terms of the age groups (Figure 7) shows that the higher age groups, concerning lower age individuals, believe that the mining activities had the greatest impacts on the environmental and health indicators. The mean of cultural outcomes related to mining activities at the lower age groups is greater concerning other age groups. On the other hand, the age groups between 10 and 20 years, and also higher than 50 years, believe that the presence of a mine close to their village not only had not caused economic productivity for them but it had a negative impact on their economy.

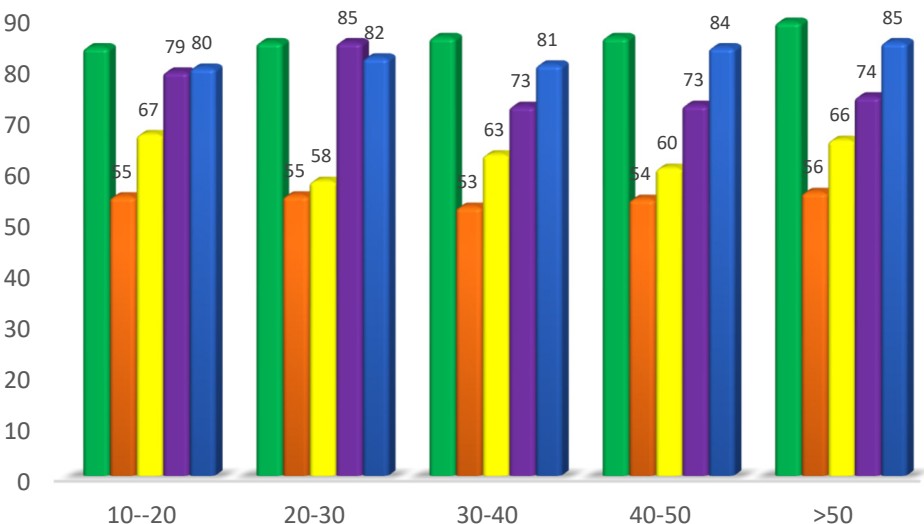

**Figure 7.** Comparison between the mean scores of sustainable development indicators in terms of the respondents' age.

Results from Table 3 show that the mining activities had the highest impact on the environment, health, and culture of the residents around the mine, from the viewpoint of respondents.

**Table 3.** Descriptive statistics of sustainable development indicators in the studied sample.

|  | N | Mean | Std. Deviation | Skewness | | Kurtosis | |
|---|---|---|---|---|---|---|---|
|  | Statistic | Statistic | Statistic | Statistic | Std. Error | Statistic | Std. Error |
| Environment impact |  | 86.32 | 7.46 | −0.50 | 0.21 | −0.84 | 0.43 |
| Economic impact |  | 54.55 | 4.44 | −0.92 | 0.21 | 0.43 | 0.43 |
| Social impact | 102 | 63.93 | 5.11 | −0.46 | 0.21 | −0.22 | 0.43 |
| Cultural impact |  | 75.82 | 17.39 | −0.64 | 0.21 | −0.40 | 0.43 |
| Health impact |  | 80.86 | 9.98 | −0.05 | 0.21 | −0.51 | 0.43 |

The skewness and kurtosis coefficients of the main variables of the research fall in the allowable interval of $(-5, 5)$ and $(-3, 3)$. In other words, a sufficient condition for the normal distribution of the data exists. It means that, while variables have a proper interval scale (have quantitative nature), they also have followed the normal bell pattern in the frequency distribution of the data, with the two sufficient and necessary conditions, and it is possible to use the parametric tests and software (Figure 8).

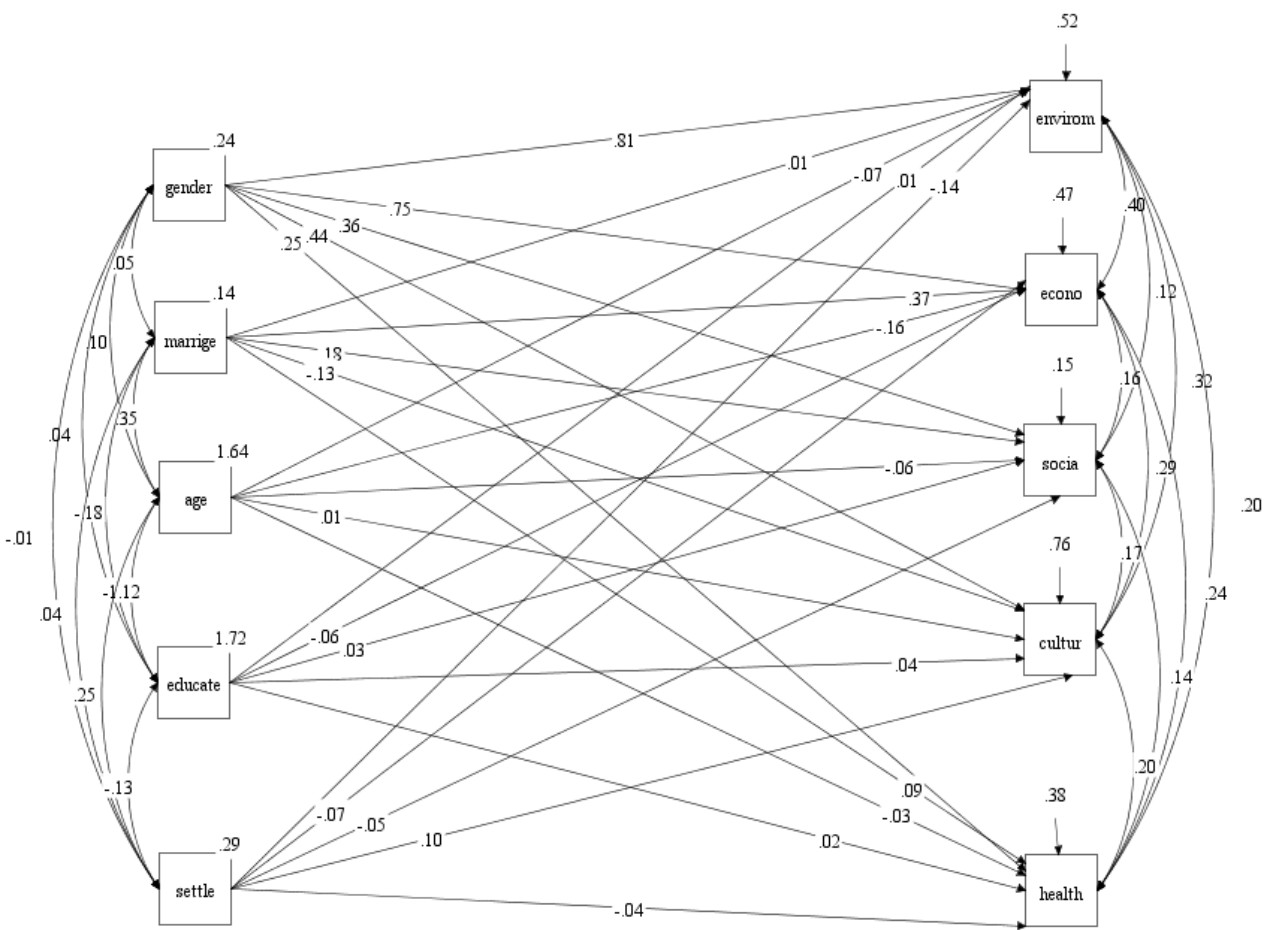

**Figure 8.** Mimic model for the state of estimating the standard coefficients.

The Multiple Indicators Multiple Causes(MIMIC) model has been formulated, following the goals initially established, predicting the behavior of the dependent variable. Model fitting indices indicate the good establishments of the MIMIC model of sustainable development indicators, where χ2/df is less than 1, the root mean square error of approximation (RMSEA) is less than 0.05, and the values of the goodness of fit index (GFI), the comparative fit index (CFI), and the Tucker–Lewis index (TLI) were greater than 0.9.

Despite the difference in the mean values of many sustainable development indicators in the sample, Table 4, only the two indicators of environmental and economic outcomes of the mining activities, in men and women groups of the studied population, have a significant difference with 95% probability. This means that the men group in the studied village significantly believed that mining activities have caused stable environmental defects, such as air, water, and soil pollution in their living area. On the other hand, the women's group believed that mining activities close to their village not only had not caused an economic boom but had even reduced the economic productivity in their living area by abandoning agricultural lands, animal husbandry, and other related activities. In addition, there was no significant difference in terms of other sustainable development indicators in the area studied.

The findings of the present research revealed that the indigenous people around the studied mining unit with any gender, age, and education composition declared that mining activity is a threatening factor for health, social, and cultural indicators, and environmental preservation. Results of this study show that, among the indicators of sustainable development, the highest impact belongs to the environmental outcomes of these activities. Most inhabitants, men 0.87 and women 0.72, believe that mining activities have extensively damaged their environment.

**Table 4.** Testing the research assumptions in terms of mining activities' impact on the sustainable development indicators.

|  | Hypothesis | β | *p*-Value | T-Value | Result |
|---|---|---|---|---|---|
| Environment impact | Age | −0.076 | 0.521 | −0.64 | Accept |
|  | Gender | 0.768 | 0.013 | 2.49 | Reject |
|  | Marriage | 0.008 | 0.977 | 0.02 | Accept |
|  | Educate | 0.009 | 0.919 | 0.1 | Accept |
|  | Residence record | −0.136 | 0.34 | −0.95 | Accept |
| Economic impact | Age | −0.164 | 0.145 | −1.459 | Accept |
|  | Gender | 0.729 | 0.01 | 2.483 | Reject |
|  | Marriage | 0.37 | 0.167 | 1.382 | Accept |
|  | Educate | −0.062 | 0.447 | −0.761 | Accept |
|  | Residence record | −0.065 | 0.631 | −0.481 | Accept |
| Social impact | Age | −0.083 | 1.304 | 0.192 | Accept |
|  | Gender | 0.242 | 1.463 | 0.143 | Accept |
|  | Marriage | 0.174 | 1.158 | 0.247 | Accept |
|  | Educate | 0.01 | 0.212 | 0.832 | Accept |
|  | Residence record | −0.049 | −0.637 | 0.524 | Accept |
| Cultural impact | Age | −0.054 | −0.378 | 0.705 | Accept |
|  | Gender | 0.064 | 0.172 | 0.863 | Accept |
|  | Marriage | −0.161 | −0.475 | 0.635 | Accept |
|  | Educate | −0.016 | −0.158 | 0.875 | Accept |
|  | Residence record | 0.089 | 0.517 | 0.605 | Accept |
| Health impact | Age | −0.035 | 0.729 | −0.346 | Accept |
|  | Gender | 0.199 | 0.451 | 0.753 | Accept |
|  | Marriage | 0.086 | 0.72 | 0.359 | Accept |
|  | Educate | 0.008 | 0.911 | 0.112 | Accept |
|  | Residence record | −0.04 | 0.743 | −0.328 | Accept |

The findings of the present study are in accordance with other studies, which also reveal that environmental outcomes are among the main results of mining activities [46,47]. Farahani and Bayazidi (2018) also investigated the environmental effects of mining around a village. Their study results revealed that 80% of the local community complained about pollution of groundwater, and rivers, and damage to the natural landscape and ecosystem of their living area [25]. Mabey et al. (2020) reported that mining plans cause the loss of agricultural lands and blow to the economic activities of the villagers living close to the mine. The findings of this study showed that about 60% of the participants believed that mining had caused damage to their lands. Also, 17% of them stated that the mine was the responsible factor for reduced soil fertility and consequently reduced native agricultural products [48].

In the study performed by Monjezi et al. (2009) using the Folchi method and summing the weighted values of environmental factors affecting the open pit mines in Iran, it was found that the most important of all the effects on the open pit Sarcheshme copper mine correspond to the negative environmental impacts including air quality, changes related to earth, plants and animals with values equal to 100, 80 and 77.6, respectively. Therefore this mine causes maximum harm to the environment [13]. The results of the field study conducted by Ogbonna et al. (2015) which was performed using a structured questionnaire and field observations showed that 72% of respondents believed that a special kind of plant species in the area has become extinct. Also, 80% of the respondents agreed that a special and rare type of an original species of animal has been lost in the mining area. The findings of the study emphasized the role of mining activities in changes in the settlement and environmental patterns of the communities [49]. In an analytic study by Yang and Ho (2018) based on the model of behavior modification theory in the mining areas, 77.3% of the inhabitants of 37 villages believed that environmental pollution is very serious. A higher percentage (85.4%) of respondents emphasized that pollution had direct impacts on

their health and coefficient R2 = 0.196 confirms that there are serious concerns over health deterioration and disease prevalence which in themselves could turn into conflicts and threats against the mines [26].

The results of the present study agree with those of Yang and Ho's (2018) study in terms of the mining activities' impact on the environmental and health indicators, but in terms of the highest impact of these activities is different from that study. In the present study, the impact of mining on the health of residents is emphasized. Near 80.6% of the participants in this study, especially the group of women, and individuals with higher education and residence records were concerned about it. The study of Mohsin et al. (2021) based on the Folchi method applied for environmental components showed that the affected population included 12 villages and according to Folchi reasoning, air pollution with 10 scores, and surface and groundwater pollution respectively with 9 and 8 scores had the highest ranks in terms of the effective factors on the health of the village residents in terms of mining activities [50]. The findings of Shoko and Mwitwa's (2015) study showed that 79% of the participants who dwelt around the studied mine had no access to healthy water. The study results showed that the mine by releasing toxic compounds such as heavy metals in the air, water, and soil causes diseases such as cancer, respiratory problems, and chronic diseases in individuals, especially in sensitive groups such as children and elders [51]. The impacts of mines on the local communities are very diverse. According to Gueye et al. (2021), the mining sector of Quebec has a considerable share in the financing of the communities. On average, it provides nearly 30,000 jobs directly or indirectly which the local workers have. So that the salary of the mine workers is higher than that of other local jobs. In 2016, the average salary in the mining sector was 1261.14 dollars and for this reason, many local inhabitants were willing to be hired for working in mines [52]. The economic outcome is accounted as another important sustainable development indicator that was examined in the present study. The results showed that the mining activities had the minimum impact and importance (54.55) on the economic situation of the communities so the individuals believed their economic status has not improved since the beginning of the mining activities. The women of the studied community had more complaints about the negative impact of mines on their family economy. They believed that the presence of the mine not only has not increased the economic productivity of their families but even has great negative impacts on them including a reduced workforce for agricultural activities and lack of access to agricultural lands leading to change in their living conditions. The study performed by Jurzina et al. (2017) showed that mining had caused the loss of fixed assets of local inhabitants which were their agricultural lands. As the result, the area under cultivation has greatly reduced and the main economic source has been greatly hurt [53]. Perhaps no other industry such as mining has caused disputes over lands, while it is expected that mining activity is an economic driver of the local communities leading to increased revenues and job opportunities. But inequalities in revenues and acquiring agricultural lands from individuals through the mines have taken away job opportunities from the communities, leading to increased poverty and the creation of tension between communities and mines. The findings of Shoko and Mwitwa (2015) and Kemp and Keenan (2010) [51,54] show that local individuals are less employed in jobs with high revenues so the employment of individuals is the driver of the economy that should be under consideration. In contrast, the results of Al Rawashdeh et al. (2016) showed that the most important positive aspect of mining was that it had created major economic growth for individuals so that from the start of mine operation the unemployment rate in the area had been reduced from 27.9% in 1990 to about 23% in the next year (2013) [55]. In the study of Pokorny et al. (2019), the results were different and distinguished in two provinces located close to the mine. In one province, more than 80% of studied families declared that their living conditions had deteriorated and believed that mines had caused a lack of access to their agricultural lands and some families had been obliged to resettle in other areas. But in the other province near this mine, 70% of the respondents declared that their life had improved at the same time, and believed that mines had caused an increase in revenues, especially for the young

people. The difference between ideas in terms of understanding the negative impacts of the two abovementioned mines shows that understanding the negative impacts of mining is related to the severity of mining and extraction [56]. The results of economic assessments by Farahani and Bayazidi (2018) showed that the economic impacts of sand and gravel mines were small on the living of the communities. Generally, employment with 75% frequency and career variety with 95% frequency have low and very low negative impacts on the inhabitants, respectively [25]. Therefore correct economic functioning should be directed to achieving efficient use of resources including the workforce, development of business, and job creation. Mining activities in addition to the economic impact on the communities, also affect the social life of individuals.

As was reasoned in this study, the concerns over the social impacts on the individuals with higher residence records in the areas around the mines are greater. The residents also had expressed concern over a variety of social infrastructure such as connecting roads and road infrastructure alongside the environmental and economic situations. In addition, the issues related to the population such as population density or abandoning the area by young families is another important aspect of social impacts that are often ignored. Mining activity has increased the percentage of participation and social interaction with corresponding authorities by 50% which has resulted in public welfare. But on the other hand, the conflicts and differences between people and authorities have increased by 90%. Also by the start of the operation of the mine, traffic has increased to its maximum rate affecting the social life of people [50]. The results of Shoko and Mwitwa (2015) studies show that about 63% of respondents believed that the presence of the mine has not resulted in the creation of infrastructure and development of the community such as roads, higher education institutes, and hospitals and a total of 37% of respondents declared that the mining companies have built a school in addition to the creation of agricultural cooperatives or renovation of them [51]. Therefore, enhancing the social welfare of local communities means fulfilling the basic needs of society. It should create equal opportunities for community participation and individual involvement. Cultural outcomes are another sustainable development indicator in mines that are assessed. The results of the present study showed that concerns over the impacts of mining on the cultural sources and the associated outcomes such as changes in the native style, architecture, and aesthetic of the area, were more felt in the lower age group which reveals that protecting the local cultural factors is very important for preserving the lifestyle in rural communities by the new inhabitants. A study was carried out by Mabey et al. (2020) [48] on the subject of the impact of mining on cultural indicators. The result of the study by Mojarradi et al. (2016) [23] using the exploratory factor analysis showed that 12% of the local communities believe that mining results in a loss of cultural identity. In other words, mining activities are a serious threat to the identity and cultural sources in an area. Protection and rehabilitation of abandoned lands after the close of the mines is a factor in preserving the cultural landscapes and heritage of the community to prevent the outcomes and defects after mining. This reasoning is also confirmed by another study performed by Lei et al. (2016) [57]. What is known is that mines as reliable sources of economy for many countries have many impacts and various outcomes for the local communities. These impacts could be investigated in terms of sustainable development programs from the two positive and negative aspects. The sustainable development programs of mines should be formulated and planned based on the specific cultural, social, political, and environmental realities of each area. Si et al. (2010) reported that according to the views of many inhabitants living around the mines, preserving the environment has a higher priority over economic interests [58]. The results of the study have some managerial implications and can help Legal organizations and mine managers to realize the effects of mining activities on sustainable development indicators.

Our findings indicate that mine managers can improve in weak areas of sustainable development indicators by improving design processes, respecting people's cultures, customs, and values, recognizing local communities as stakeholder groups, participating in the social, economic, and institutional development of local communities, providing

information and training for the safe handling, use, transport, and disposal of the materials, Researching processes, practices, and technologies that will lead to improved sustainable development indicators, etc.

## 4. Conclusions

Sustainable development is one of the main objectives of mines, but the nature of mineral activities always in a way that it leaves many negative impacts on the surrounding environment. In the present study, by reviewing the literature and interviews with experts, the indicators of sustainable development were identified and categorized. Then in the form of a standard questionnaire, the relevant data were collected from the target statistical sample. The study results showed that mineral activities had the highest impact on environmental indicators and individuals' health. Also, it has a minimum impact on financial and economic satisfaction. Furthermore, the group of men in the studied population significantly showed the highest concern over the sustainable environmental defects in the mines. In contrast to the men, the women believed that the presence of mines close to their residential areas had an impact on their economic situation. The findings of the current study reveal that most of the residents close to the mine site have expressed their concern over the impact of mining activities on the environment and their health. Thus, mines should define and implement objectives and formulate plans based on the realities of their covered areas. The present study, such as many similar studies, has a few limitations including the small sample size in terms of both quantitative and qualitative sections. Albeit a low sample size is among the intrinsic limitations of qualitative studies, future studies could study sustainable development indicators using large sample sizes. The other limitation of the present study is that the presented model could be generalized only for open-pit type mines in Iran but in future studies, it could be expanded to include other types of mines such as underground ones. Furthermore, the number of predictor variables in the presented model is four, which could be increased in future models.

**Author Contributions:** Methodology, H.D. and A.A.K.; Software, H.D. and K.A.F.; Validation, H.D., Marc Bascompta and A.A.K.; Formal analysis, H.D., A.A.K. and K.A.F.; Investigation, H.D., A.A.K. and K.A.F.; Data curation, K.A.F.; Writing—original draft, M.B.; Supervision, M.B. All authors have read and agreed to the published version of the manuscript.

**Funding:** This study was supported by the Hamedan University of Medical Sciences and Health and ethical code IR.UMSHA.REC.1400.617. The study program was approved by the national committee for ethics in medical research (IR.UMSHA.REC.1400.617).

**Informed Consent Statement:** Not applicable.

**Data Availability Statement:** Not applicable.

**Conflicts of Interest:** The authors declare no conflict of interest.

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
