# Peer review of "A Mimic Model Approach for Impact Assessment of Mining Activities on Sustainable Development Indicators"

_sustainability, doi:10.3390/su15032688_

Round 1
Reviewer 1 Report
Comments to the text
1. Errors noticed:
- page 3- title of subsection 2-1 should be: Qualitative study
- fig.1: by the „open coding) complete or/and remove bracket
- tab 1: bachelors percentage have to be 8% (not 0,08%)
- fig 5: lack of education groups explanation
Comments
1.Presented way of codification and sustainable development identification indicators (categories) seems to be interesting, hovever such indicators as well as indicators of mining impact are well documented and described in mamy papers.
2. The representativeness of the sample of respondents and its size should be discussed/ described. This also applies to the place/region where the survey was conducted.
Author Response
we appreciate the constructive comments of the respected reviewer. The following are our point-by-point responses:
Point 1: - page 3- title of subsection 2-1 should be: Qualitative study
Response 1: Thank you for this point. It was done.
Point 2- fig.1: by the „open coding) complete or/and remove bracket
Response 2: Thank you for this point. It was done.
Point 3- tab 1: bachelor’s percentage have to be 8% (not 0,08%)
Response 3: Thank you for this point. Table1 was revised.
Point 4- fig 5: lack of education groups explanation
Response 4: Thank you for this point. It was done.
Point 5: Presented way of codification and sustainable development identification indicators (categories) seems to be interesting, however such indicators as well as indicators of mining impact are well documented and described in many papers.
Response5: Thank you once again for your valuable comments and suggestions. Identification and classification of sustainable development indicators in the present study is based on literature review and interviews with experts in the form of a qualitative study. This approach greatly affects the results of the quantitative part of the study. The data collection tool (questionnaire) is based on the primary codes extracted from the qualitative part of the study.
Point6:The representativeness of the sample of respondents and its size should be discussed/ described.
Response6: Thanks for raising this important point. It was done (lines (155-160)
Again, we appreciate all your insightful comments. We worked hard to be responsive to them. Thank you for taking the time and energy to help us improve the paper

Reviewer 2 Report
This manuscript investigated the outcomes of mining activities and their testing in one of the open pit mines of Iran. The outcomes by the mining company were assessed in order to achieve a proper management. The topic of this paper falls within the scope of “Sustainability” and is of topical importance in the field of sustainable development of mines. This work contributes to formulate the controlling strategies for potential negative outcomes of mining and achieve an effective sustainable development. I would suggest the manuscript to be accepted after some minor revisions. Please check the attachment.

Author Response
we appreciate the constructive comments of the respected reviewer. The following are our point-by-point responses:
Point1: What’s the paper highlight? What new information does the paper offer? And summary part
should be more concise
Response1: Thank you once again for your valuable comments and suggestions. Based on this suggestion - The introduction has been revised (focusing on the background of research, the motivation of the research, and the novelty) and the literature review.
- the contribution of study are explained in the introduction section
- We have added managerial insights to the end of discussion section.
Point2: I strongly recommend the authors have a second look at the article, correct the grammatical
errors and avoid the minor mistakes and non-standards
Response2: Thank you for all of your detailed comments and suggestions.
- The full text of the article was revised by a professional editor
Point3: The abstract and the conclusions should include some of the key data such as the difference between men and women in terms of the environmental and economic indicator
Response3: Thank you for all of your detailed comments and suggestions. It was done(lines(22-23),(459-468).
point4: The research background is not well summarized, so the “Introduction” of this manuscript should be reorganized. Flotation process in mining company will affect the mining activities, so
it should be described in the “Introduction”, and several relevant references may be added to
support this point, such as Int. J. Min. Sci. Technol. 32 (2022) 1351–1364; Miner. Eng. 187
(2022) 107796; Sep. Purif. Technol. 307 (2023) 12277
Response4: Thank you for this point. It was done. References list was revised
Point5: The figures are poorly presented, such as Figures 1 and 8, and the author should revise the format carefully.
Response5: Thanks for raising this important point. Figures 1 and 8 were revised
Point6: There is a mixed explanation of the methods in results and discussion, these must be in the methods section
Response6:Thank you for this valuable comment. The explanations related to the mixed method were removed from the discussion and introduction sections. Explanations were added at the beginning of the methods section.
Point7: Some examples of editing or writing issues: “... it is crucial to identify and assess
these outcomes by the mining company in order to achive a proper management”, “...
the average weakly salary in the mining sector was 1261.14 dollars and for this reason
many local inhabitants were willing to be hired for working in mines”.
Response7: This is another great point. The manuscript was revised again by an experienced English language editor
Again, we appreciate all your insightful comments. We worked hard to be responsive to them. Thank you for taking the time and energy to help us improve the paper

Reviewer 3 Report
The paper is good but should be revised. Figure 1 is not suitable for the present journal, the authors captured the graphic. Please use the file in eps format. The present format is poor, Figure 1 should be improved. What are the contributions of the paper, the Introduction is very long but is just a literature review, we request the authors to add the contributions and the novelties of the present paper. Please give the source of the cloud diagram in Figure 3. Is it possible to get the units of CVR and CVI? The number of observations for the econometrics study should be given. The specification test of the model used should be given. Is it possible to get the R square obtained in your estimations? The paper needs minor revisions. Is possible to get the results with Stata, please give explanations.
Author Response
we appreciate the constructive comments of the respected reviewer. The following are our point-by-point responses:
Point1: Figure 1 is not suitable for the present journal, the authors captured the graphic. Please use the file in eps format. The present format is poor, Figure 1 should be improved.
Response1: Thank you for this valuable comment, Figure has been revised
Point2: What are the contributions of the paper, the Introduction is very long but is just a literature review, we request the authors to add the contributions and the novelties of the present paper.
Response2: Thank you for this point. In the revision, The introduction has been revised
- the contribution of study are explained in the introduction section
- We have added managerial insights to the end of discussion section.
Point3: Please give the source of the cloud diagram in Figure 3.
Response 3 :Thank you for this point. In the revision, the source of the cloud diagram was added in the caption section.
Point4: Is it possible to get the units of CVR and CVI? The number of observations for the econometrics study should be given.
Response4: Thank you for this direction. A review of the content validity literature shows that the indices CVR and CVI do not have units. Explanations related to CVR index calculation were added.
Point5: The specification test of the model used should be given. Is it possible to get the R square obtained in your estimations? The paper needs minor revisions. Is possible to get the results with Stata, please give explanations.
Response5: Thank you for all of your detailed comments and suggestions.The specification test of the model was added(lines 289-293). Mimic models are a special type of structural equation modeling that is implemented with the purpose of testing the impact of multivariates(age, level of education,gender) on a measurement model. In these models, unlike causal models, usually a scale variable is measured among several covariates. However, because the additive composite in the MIMIC model incorporates all of the variation from the formative indicators (R2 = 1). Yes, it is possible to model with Stata and other programs such as SAS CALIS procedure and R lavaan package. Mplus is designed primarily for latent variable modeling and has far more modeling flexibility compared to SAS and R, stata
Again, we appreciate all your insightful comments. We worked hard to be responsive to them. Thank you for taking the time and energy to help us improve the paper

Reviewer 4 Report
It is an interesting work that will surely be applied in the field of study knowledge, the study methodology could be applied to other areas of industrial activity.
Author Response
we appreciate the constructive comments of the respected reviewer. The following are our point-by-point responses:
The manuscript was revised again by an experienced English language editor.
Again, we appreciate all your insightful comments. We worked hard to be responsive to them. Thank you for taking the time and energy to help us improve the paper
